# Chemical Analysis of Plasma-Activated Culture Media by Ion Chromatography

**DOI:** 10.3390/ph18020199

**Published:** 2025-02-01

**Authors:** Marcello Locatelli, Miryam Perrucci, Marwa Balaha, Tirtha-Raj Acharya, Nagendra-Kumar Kaushik, Eun-Ha Choi, Monica Rapino, Vittoria Perrotti

**Affiliations:** 1Department of Science, University “G. d’Annunzio” of Chieti-Pescara, 66100 Chieti, Italy; 2UdA-TechLab, Research Center, University “G. d’Annunzio” of Chieti-Pescara, 66100 Chieti, Italy; 3Department of Biosciences and Agro-Food and Environmental Technologies, University of Teramo, 64100 Teramo, Italy; mperrucci@unite.it; 4Department of Innovative Technologies in Medicine & Dentistry, University “G. d’Annunzio” of Chieti-Pescara, 66100 Chieti, Italy; 5Department of Pharmacy, University “G. d’Annunzio” of Chieti-Pescara, 66100 Chieti, Italy; marwa.balaha@unich.it; 6Department of Pharmaceutical Chemistry, Faculty of Pharmacy, Kafrelsheikh University, Kafr El Sheikh 33516, Egypt; 7Plasma Bioscience Research Center, Department of Electrical and Biological Physics, Kwangwoon University, Seoul 01897, Republic of Korea; tirtharajacharya2050@gmail.com (T.-R.A.); kaushik.nagendra@kw.ac.kr (N.-K.K.); ehchoi@kw.ac.kr (E.-H.C.); 8Unit of Chieti, Genetic Molecular Institute of CNR, University “G. d’Annunzio” of Chieti-Pescara, 66100 Chieti, Italy; monica.rapino@unich.it

**Keywords:** anionic chemical profile, analytical method, ion chromatography, pharmacokinetics, plasma jet, RONS analysis

## Abstract

**Background**: Currently, the procedures and methods applied in biological and medical fields for the determination of reactive oxygen and nitrogen species (RONS), primarily rely on spectrophotometric techniques, which involve the use of colorimetric reagents. While these methods are widely accepted, they exhibit significant limitations from an analytical standpoint, particularly due to potential inaccuracies, artifacts, and pronounced susceptibility to matrix effects. The purpose of this Technical Note is to demonstrate the application of ion chromatography—a robust and well-established analytical technique—for the quantification of RONS produced in cell culture media through the exposure to cold atmospheric plasma (CAP), an innovative therapeutic approach for cancer treatment, known as CAP indirect treatment. In addition, the present protocol proposes to apply the pharmacokinetics principles to the RONS generated in plasma-treated liquids (PTLs) following CAP exposure. **Methods**: The strategy involves elucidating the kinetic profiles of certain characteristic species by evaluating their half-life in the specific media used for cell cultures and investigating their “pharmacokinetic” (PK) profile. In this approach the drug dose is represented by the plasma power and the infusion time corresponds to the exposure time of the culture medium to CAP. Volume-dependent results were shown, focusing on nitrites and nitrates activities, justifying cellular inhibition. **Results**: This methodology enables the correlation of the PTL biological effects on different cell lines with the PK profiles (dose/time) obtained via ion chromatography. **Conclusions**: In conclusion, being a simple and green method, it could be used as an alternative to toxic reactions and analytical techniques with higher detection limits, while achieving good resolution.

## 1. Introduction

Cold atmospheric plasma (CAP) is an ionized gas that operates at ambient pressure and room temperature (<40 °C). It is composed of free electrons, ions, radicals, and reactive species (e.g., reactive oxygen and nitrogen species—RONS), as well as excited atoms and molecules, neutral molecules, electromagnetic fields, and UV/Vis radiation [1,2]. There are two recognized approaches for CAP application:(i)Direct plasma treatment: This involves the direct exposure of biological targets (e.g., cells, tissues, or animals) to CAP, resulting in a synergistic effect of all the plasma components mentioned above.(ii)Indirect plasma treatment: In this method, a liquid (e.g., cell culture medium) is exposed to CAP and then applied to the biological target. In this case, the effect is mainly due to the RONS generated in the activated liquid (PTL: plasma-treated liquid), particularly through its enrichment with long-lived secondary RONS [3].

Among the liquids that can be exposed to CAP, various water-based solutions commonly used in clinics—such as Ringer’s lactate solution, phosphate-buffered saline, deionized water, and electrolyte rehydrating III solution—show potential for clinical application of this emerging technology [3,4].

CAP demonstrated remarkable anti-tumoral potential across a wide range of cancer types [5], although its underlying mechanisms remain unclear. Several short-lived species (e.g., radical hydroxyl OH, anion superoxide O_2_^−^, and singlet oxygen ^1^O_2_) and long-lived species (hydrogen peroxide H_2_O_2_, nitrite NO_2_^−^, and nitrate NO_3_^−^) have been detected in both gas and liquid phases [6]. Hence, the standardization of chemical analyses for plasma-generated species in liquids is an urgent priority and could play a key role in elucidating the role of individual species in eliciting specific biological effects.

Nowadays, the chemical characterization of PTLs primarily involves the quantification of NO, H_2_O_2_, and total ROS using commercial test kits. These kits typically reply on UV/Vis or fluorescence (FL) spectroscopy to detect single species. The Griess test detects stable metabolites of nitric oxide (NO), such as nitrites and nitrates, through a colorimetric assay with UV/Vis absorbance at 540 nm. It involves the reaction with sulfanilic acid and *N*-(1-naphthyl)-ethylene diamine hydrochloride to produce a pink compound. Other methods include the QuantiChrom Nitric Oxide Assay Kit (BioAssays Systems, Hayward, CA, USA), the Measure-IT Nitrite assay, based on the reaction of 2,3-diaminonaphtalene (DAN) with nitrite under acidic conditions to generate 1*H*-naphthotriazole, and reactions with 2,6-dimethylphenol for nitrate detection. Graves D.B., 2014 [7], highlighted the potential role of NO biochemistry in plasma oncology, particularly noting the formation of peroxynitrite (ONOO^−^) under specific conditions, which can be detected using 2′,7′-dichlorodihydrofluorescein (DCDHF).

For ROS quantification, 2′,7′-dichlorodihydrofluorescein diacetate (DCFH-DA) assays are commonly employed. H_2_O_2_ detection often relies on the Amplex™ Red Hydrogen Peroxide/Peroxidase Assay Kit (Thermo Fisher, Waltham, MA, USA) or QuantiChrom^TM^ Peroxide Assay Kit (DIOX-250) (BioAssays Systems, USA), as well as the titanyl sulfate assay. For hydroxyl radicals (OH), the 3′-(p-aminophenyl) fluorescein (APF) assay is typically used [8]. While these techniques are widely utilized, they suffer from several drawbacks, including interference between the assay components and PTL, challenges in data interpretation across laboratories due to varying kits and equipment, and the inability to analyze multiple species simultaneously. Finally, it is worth noting that, nowadays, there is a tendency to avoid and/or minimize the use of toxic solvents, as recommended by the principles of Green Chemistry (GC) [9]. To address these limitations, ion chromatography (IC) presents a promising alternative. IC, a form of liquid chromatography based on ion interactions with a stationary phase made of ion-exchange resins, is widely applied in the analysis of aqueous samples containing common anions (i.e., fluoride, chloride, nitrite, nitrate, and sulfate) [10]. To the best of our knowledge, IC has not yet been applied to PTL characterization. It is most employed for the quantification of anions in water quality analysis, environmental monitoring, pharmaceutical compounds and food/beverage analysis, biomedical research, and chemical process control. IC offers several advantages, including the ability to simultaneously analyze multiple species (multianalyte analysis), resulting in fast, accurate, and reproducible results. In addition, it enables the correlation of anion quantitative data with PK profiles using computational tools such as Excel Add On PK Solver, an extension of Excel (Microsoft Office) made of different methods, as non-compartmental or compartmental analysis [11].

Therefore, this article aims to provide a detailed protocol to standardize the chemical analysis of PTLs and facilitate their potential clinical application as therapeutic agents. The protocol has already been applied to plasma-activated cell culture media, enabling the investigation of their effects on cell proliferation, apoptosis occurrence, and cell cycle regulation in a panel of head and neck cancer (HNC) cell lines [12]. This approach allows for the selection of the most effective PAM for therapeutic purposes.

## 2. Results and Discussion

### 2.1. Analysis of Results

Once areas are obtained from Chromeleon, a calibration curve was used to correlate them with the corresponding concentrations, reported in µg/mL. With the Excel Add On PK Solver, the PK of the abovementioned anions was calculated, taking into account the following parameters:

Type of administration;

Dose;

Time of infusion;

Time on abscissa and concentration on ordinate.

The type of administration was set as non-compartmental IV constant infusion (NCA IV infusion); the dose considered was the device’s duty cycle and power calculated as follows.

The duty cycle can be defined as [13,14]Duty cycle (%) = (On time)/(On time + Off time) × 100(1)Energy (E) = Q × V= ∫_t1^t2V(t)I(t)dt [J](2)Power (P) = Duty cycle × E × 1/t [W](3)
where Q = charge, V = peak voltage, I = peak current, and t = plasma discharge time.

Time of infusion was set as the time of sample treatment (5, 10, or 20 min exposure time). With the non-compartmental method, several parameters, fundamental for PK assessment, are obtained:✓Lambda_z (individual estimate of the terminal elimination rate constant);✓T½ (half-life);✓Tmax (time when the maximum concentration is found);✓Cmax (maximum concentration found);✓Clast_obs/Cmax (the observed concentration at Tlast, where Tlast is the time of the last sample measurable with the analytical method);✓AUC0-t (expressed in µg/mL × min);✓AUC0-inf_obs (expressed in µg/mL × min);✓AUC0-t/0-inf_obs (expressed in µg/mL × min);✓AUMC0-inf_obs (the area under the moment curve from time zero (pre-dose) to the time of the last sample, including after the application of the LOQ rules);✓MRT0-inf_obs (the mean residence times, based on AUC);✓Vz_obs (apparent volume of distribution) expressed in (J/s)/(µg/mL);✓Cl_obs (systemic clearance) expressed in (J/s)/(µg/mL)/min;✓Vss_obs (volume of distribution at steady state) expressed in (J/s)/(µg/mL).

### 2.2. Results

Given the growing interest in using CAP as a potential treatment for cancer, it was deemed important to evaluate the PK profiles of analytes that are considered critical in this field. The interest in CAP’s efficacy against cancer is rooted in the differences in oxidative states between normal and tumor cells. Specifically, cancer cells have a higher concentration of ROS, lower antioxidant activity, and greater porosity. Based on this evaluation, our aim was to understand how the chemical composition of PTL could affect cell viability. The 6 mm distance was chosen as the best condition between the plasma plume and the media’s surface through a deep revision of the literature and considering 6 mm as an average. At a close distance of 3 mm, the plasma jet is dense, leading to a dominance of nitrogen-based species (RNS) such as NO_2_^−^, due to strong interactions with nitrogen molecules. As the distance increases to 6 mm, the plasma plume expands, creating a balanced interaction between nitrogen and oxygen, resulting in a mixture of both RNS and ROS, including H_2_O_2_ and superoxide. At longer distances (9 mm), the plasma disperses further, reducing nitrogen interactions and increasing the concentration of oxygen-based species (ROS), such as H_2_O_2_ and superoxide.

In this way, the safety of the operator is ensured, but media were mainly subjected to uniform treatment, avoiding undesirable non-uniform effects due to being too close to the instrument, and plasma plume was protected from possible drops generated by collusion between flame and media. In addition, temperature, with this distance, appears more stable, avoiding heating and protecting heat-labile compounds.

Therefore, the present protocol aims to provide selective, accurate, rapid, and sensitive quantification of the anions produced in PTLs, which are considered fundamental for the anticancer effects of indirect CAP treatment. It is important to note the use of high-purity water (HPW), as it does not contain anions that could interfere with the analysis. Similarly, the use of transparent rack and Falcon tubes is crucial for visualizing the plasma plume and the surface of the media. This method was fully validated in accordance with ICH Guidelines [15,16,17].

Firstly, we ensured that the blank matrix (i.e., medium without plasma activation) did not contain the analytes of interest. If any analyte was detected in the blank matrix, its values were subtracted from the subsequent quantification to avoid skewing the results, as shown in Figure 1. The absence of anions of interest in the blank matrices is evident, as also supported by the presence of tangent lines in their respective retention times (considering also the standard deviation stdev), when doubts were present. To ensure the absence of overlap regarding the first peak, a dilution 1:1000 was carried out for each matrix, obtaining a single peak, which was useless for this study.

Table 1 reports the major analytical parameters used for the quantification of each anion. The sensitivity of this method is of crucial importance, as it allows for the quantification of anions even at very low concentrations.

It is worth noting that in a previously published study [12], IC allowed us to understand how the volume-dependent proliferation inhibition in HNC cells was linked to the volume-dependent increase in nitrites and nitrates. A similar result was obtained by Zhang et al., which confirmed data present in the literature showing that the increase in nitrites and nitrates correlate with a decrease in cell proliferation [18].

The key element of this protocol is that it describes, for the first time, a robust analytical method to standardize the chemical analysis of PTL administrated as a drug, following typical principles of pharmacotoxicology (evaluation of PK profiles with experimental parameters). In this way, plasma can be considered the “drug” and the culture medium, the “patient”, analyzing the latter’s response to the treatment, which was administered on the first day. Thus, using Excel, we obtained the PK parameters, as previously shown [11]. It was interesting to observe that nitrite and nitrate concentrations increased in a volume-dependent manner, with their levels rising until X hours and then decreasing at the final time point (72 h). Unlike nitrites and nitrates, no scientific studies in the literature have highlighted the importance of phosphate and sulfateSome PK parameters obtained from this study are reported in Appendix A, as also previously shown in our last publication [12]. Additionally, some chromatograms are shown in Appendix A.

Furthermore, the protocol described here refers to an experimental setting where 2 and 5 mL of culture medium were activated for 5, 10, and 20 min, maintaining a 6 mm distance between the tip of the plasma jet and the media. However, it can be easily extended to any other experimental setup, allowing PTL to be characterized and administered as a drug in a unique and reproducible way. The simplicity of the protocol, the absence of toxic/hazardous substances, and the type of instrumentation make this method applicable not only to other PK studies and potential PTL applications but also to further research related to the parameters mentioned above.

Additionally, PAM is useful also for other pathologies, as reported by Martin and Yao, who discussed its role in inhibiting fungal and viral growth [19,20].

Based on the easy use of this protocol, it is important to also highlight the greenness of the method, which does not use organic solvents as mobile phases and is able to detect a low concentration of ten analytes; thus, it could be defined as a multianalyte method, as required by Green Analytical Chemistry principles.

## 3. Materials and Methods

### 3.1. Chemicals and Materials

Multi Anion Standard 1 for IC, containing fluoride, chloride, bromide, nitrate, sulfate, phosphate, chlorite, bromate, nitrite, and chlorate, was purchased from Supelco (Milan, Italy). Sodium carbonate (Na_2_CO_3_) and sodium bicarbonate were obtained from Sigma Aldrich (Milan, Italy); meanwhile, PTFE filters were sourced from Whatman, (Clifton, NJ, USA). McCOY’s 5, DMEM, and RPMI were bought from Gibco (Waltham, MA, USA). Double-distilled water was obtained using a Millipore Milli-Q Plus water treatment system (Millipore Bedford Corp., Bedford, MA, USA).

### 3.2. Equipment

For the cold plasma air jet, a voltage controller was used to regulate the peak voltage, and the feed air gas flow rate was maintained at a steady 2 L per minute (Lpm). The high-voltage electrode was made of stainless steel, measuring 1.20 mm by 0.27 mm, and served as the inner electrode. The ground electrode was also made of stainless steel, with dimensions of 6.00 mm in length and 0.27 mm in thickness and a centrally drilled hole of 0.70 mm for plasma formation.

For the analytical part, a Dionex ICS 1600 (Thermo Fisher, USA) was equipped with an AS autosampler with controlled temperature and a DS6 heated conductivity cell detector. Chromeleon Software version 6.8 (Thermo Fisher, Waltham, MA, USA) was used for data acquisition and elaboration. During the analysis, samples were maintained at 25 ± 1 °C. The stationary phase used for the separation was a Dionex IonPac AS9-HC column (4 × 250 mm), with the temperature controlled at 30 ± 1 °C using a column oven. A Dionex AERS500 carbonate (4 mm) electrolytically regenerated suppressor, with an electrical current of 58 mA, was used during the analyses. The DS6 heated conductivity cell detector was thermostated at 30 ± 1 °C with autozero enabled.

A schematic representation of the instrument is reported in Figure 2.

### 3.3. Standard Solution Preparation

The stock solutions (SSs) of ten anion standards were prepared in mobile phase, consisting of 9 mM sodium carbonate, for which their concentrations are reported in Table 2. The working solutions used for the validation procedure were obtained by diluting the stock solution (SS) in volumetric flasks using mobile phase. The SS was stable at room temperature for three days; meanwhile, at +4 °C, the SS remained stable for longer (ten days). Based on this, it is preferrable to store the SS in the fridge.

### 3.4. Sample Preparation

A volume of 2 or 5 mL of the chosen medium was placed in transparent tubes held by a transparent rack. Transparency was essential to ensure the visibility of both the medium and the plasma plume. For the selected protocol, the plasma plume of the device was positioned 6 mm away from the surface of the media. Then, the device was switched on, directing the plasma flow perpendicularly to the center of the medium’s surface. A timer was used to measure the exposure time of each medium sample to the plasma. At the end of the designed exposure time (5, 10, 20 min), the device was turned off, and the activated medium was transferred into a vial for IC analysis. The samples were then injected into the IC system and subsequently kept at 37 °C until the next analysis.

Figure 3 presents a schematic representation of the sample treatment process, including the volumes and exposure times used in this study.

### 3.5. IC Analysis

To separate 10 anions, a 9 mM solution with Na_2_CO_3_ in MilliQ water was prepared. After sonication, the chromatographic system and the computer were turned on, whereas the “Chromeleon” software was used for the separation and quantification of the samples. Once the system was conditioned to minimize the noise floor, the analysis began. Immediately following the sample treatment, as described in Section 3.2, the solution was filtered and injected, without requiring dilution (as shown in Figure 1, where blank matrix are reported). Treated samples were reanalyzed after 24, 48, and 72 h (without any further treatment) to investigate the analytes’ profile. During this time, samples were stored at 37 °C.

For long-term storage (over one week), the instrument and column were washed with a solution containing 100 mM sodium bicarbonate, while the suppressor was turned off. Subsequently, the suppressor was cleaned with HPW. Figure 4 provides a simplified illustration of the protocol and the corresponding chromatogram.

### 3.6. Software

Chromeleon was utilized for the analysis. This software operates based on specific parameters and, upon the completion of the chromatographic run, generates a chromatogram displaying the peaks corresponding to the analytes present in the analysis. These peaks are converted into concentrations (µg/mL) using a calibration curve derived from the analysis of solutions with varying analyte concentrations. For the calibration curve and statistical analysis, GraphPad Prism 9.0 Software (GraphPad Software, Inc., San Diego, CA, USA) was employed. Specifically, experiments were repeated 3 times, with the results reported as averages and standard deviations. A two-way ANOVA test was performed for statistical evaluation.

## 4. Conclusions

This study highlights the versatility and potential of IC as a key analytical tool for the chemical characterization of PTL. Its applicability extends to various experimental conditions, including variations in CAP power, distances, volumes, and exposure times, which are likely to yield further meaningful insights. IC enables the simultaneous quantification of nitrite, nitrate, and other anions, establishing a linear correlation between biological responses and chemical composition.

Traditional methods for quantifying anion species often rely on complex or toxic chemical reactions and can be time-consuming, leading to inefficiencies. Moreover, spectrophotometric methods, while widely used, lack the sensitivity and accuracy that liquid chromatography offers. With IC, a single chromatographic run allows for the detection and quantification of 10 different anions, providing a comprehensive overview of the chemical landscape. This holistic approach has the potential to significantly enhance the understanding of biological effects and improve the reproducibility and relevance of experimental results.

## Figures and Tables

**Figure 1 pharmaceuticals-18-00199-f001:**
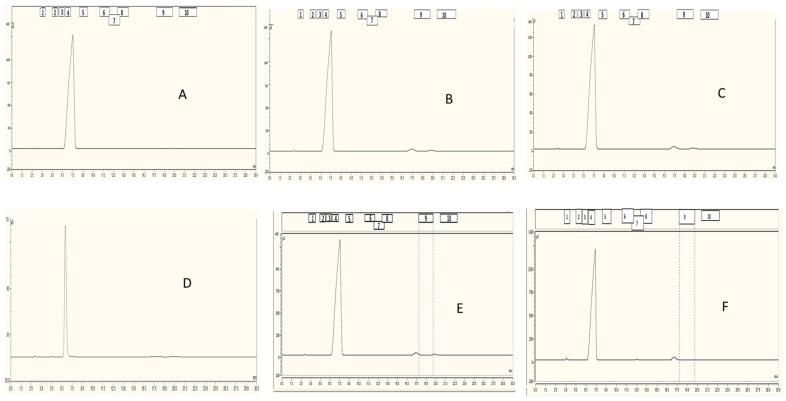
Chromatogram obtained in analyzing IC media without plasma activation. (**A**): DMEM 1:1; (**B**): McCOY 1:1; (**C**): RPMI 1:1; (**D**): DMEM 1:1000; (**E**): McCOY considering retention time (and stdev) of analytes of interest; (**F**): RPMI considering retention time (and stdev) of analytes of interest; the rectangles represent each analyte= 1: fluoride, 2: chlorite; 3: bromate, 4: chloride, 5: nitrite, 6: bromide, 7: chlorate, 8: nitrate, 9: phosphate, 10: sulfate.

**Figure 2 pharmaceuticals-18-00199-f002:**
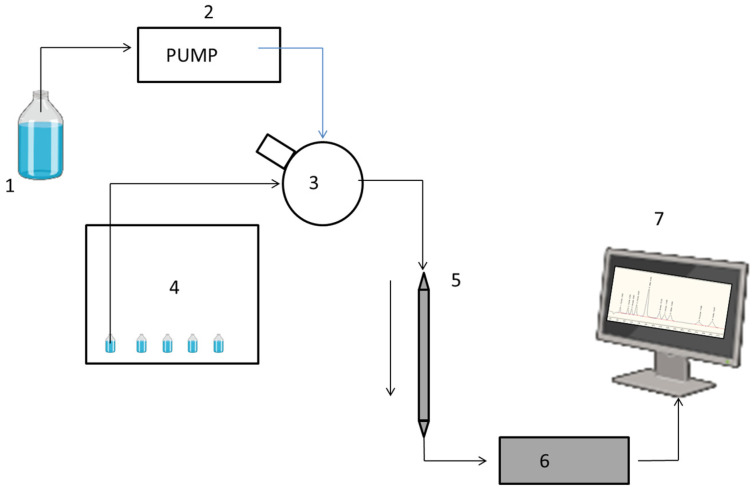
Representation of the instrument used: 1: mobile phase; 2: pump that picks up and pushes mobile phase; 3: injection valve; 4: autosampler with vials containing samples; 5: stationary phase; 6: conductivity detector; 7: computer with Chromeleon.

**Figure 3 pharmaceuticals-18-00199-f003:**
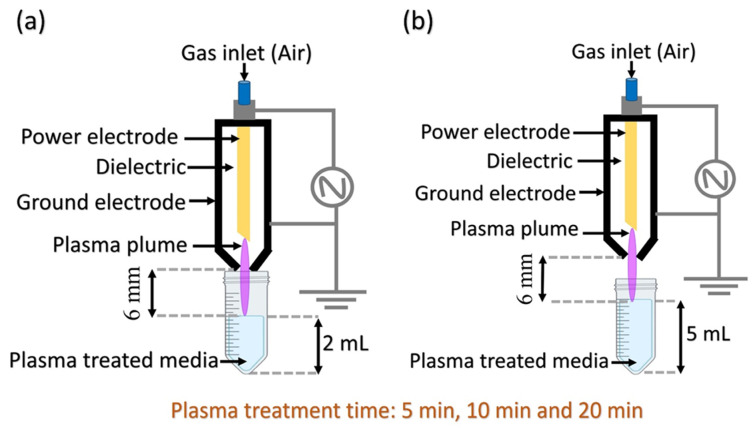
Simplified illustration of plasma-treated liquid (PTL) production using (**a**) 2 mL and (**b**) 5 mL of media.

**Figure 4 pharmaceuticals-18-00199-f004:**
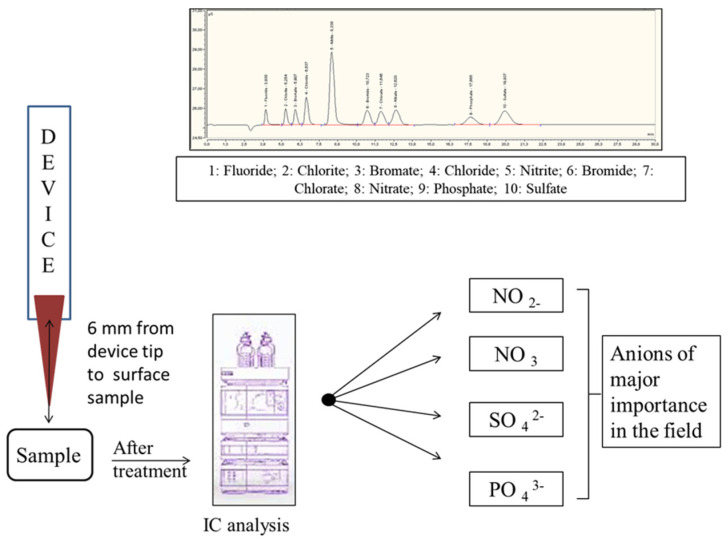
Schematic representation of the protocol reported and the chromatogram obtained from analysis.

**Table 1 pharmaceuticals-18-00199-t001:** Analytical parameters used for anion quantification.

Analyte	Slope	Intercept	Linearity ^a^	Weighting Factor	r^2^	LOD	LOQ
Fluoride	0.1266	0.003549	0.0275–2.7	1/X^2^	0.9985	0.001	0.0275
Chlorite	0.04068	0.000029236	0.101–10		0.9973	0.034	0.1010
Bromate	0.02216	0.000046963	0.202–20		0.9984	0.067	0.2020
Chloride	0.1045	0.01663	0.091–9		0.9930	0.030	0.0910
Nitrite	0.2199	0.0006067	0.152–15		0.9997	0.050	0.1520
Bromide	0.0003999	0.0002064	0.182–18		0.9989	0.060	0.1820
Chlorate	0.03462	0.0006250	0.202–20		0.9982	0.067	0.2020
Nitrate	0.04768	0.003628	0.182–18		0.9994	0.060	0.1820
Phosphate	0.02101	0.0007420	0.273–27		0.9985	0.011	0.2730
Sulfate	0.06802	0.002854	0.182–18		0.9976	0.060	0.1820

^a^ expressed as µg/mL.

**Table 2 pharmaceuticals-18-00199-t002:** Concentration of each anion calculated for SS.

Analyte	Fl^−^	ClO_2_^−^	BrO_3_^−^	Cl^−^	NO_2_^−^	Br^−^	ClO_3_^−^	NO_3_^−^	PO_4_^−^	SO_4_^−^
Conc [µg/mL]	2.7	10	18	9	15	18	20	18	27	18

## Data Availability

Data is contained within the article.

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
