# Peer review of "Chemical Analysis of Plasma-Activated Culture Media by Ion Chromatography"

_pharmaceuticals, 2025, doi:10.3390/ph18020199_

Round 1

Reviewer 1 Report

Comments and Suggestions for Authors

Locatelli et al. reported an ion chromatographic method for chemical analysis of plasma-activated culture media by applying the principles of pharmacokinetics to the RONS generated in plasma treated liquids (PTL) by CAP exposure. The present manuscript can be published in the ''Pharmaceuticals'' after minor revisions.

A conclusion paragraph should be provided for this study.

Some minor typo errors should be corrected in the manuscript. Please refer to the attached file.

The results obtained in this study should be supported with more references. Also, a comparison should be carried out with the literature to highlight the obtained data and novelty.

Author Response

COMMENT: Locatelli et al. reported an ion chromatographic method for chemical analysis of plasma-activated culture media by applying the principles of pharmacokinetics to the RONS generated in plasma treated liquids (PTL) by CAP exposure. The present manuscript can be published in the ''Pharmaceuticals'' after minor revisions.

RESPONSE: Thank you for the good introduction. We will implement all of your recommendations.

COMMENT: A conclusion paragraph should be provided for this study.

RESPONSE: A conclusion paragraph has been added as requested (page 10/12).

COMMENT: Some minor typo errors should be corrected in the manuscript. Please refer to the attached file.

RESPONSE: We deeply revised and corrected throughout the manuscript.

COMMENT: The results obtained in this study should be supported with more references. Also, a comparison should be carried out with the literature to highlight the obtained data and novelty.

 RESPONSE: Results, discussion and conclusion paragraphs have been improved and supported by further references, as suggested (paragraph 2.2 and 4, page 5/12 and 1/12).

Reviewer 2 Report

Comments and Suggestions for Authors

The technical note entitled: Chemical analysis of plasma-activated culture media by ion 2 chromatography is well written, solid article which may be interest for Pharmaceuticals Journal readers.

There are a few minor suggestions to be reconsidered, namely:

1. Please add section described statistical tools and NCA software.

2.  Please define validation procedures, including applied guidelines.

3. Please add a conclusion section. A small discussion will be beneficial.

To sum up, I reccomend the publication after reconsideration of suggestions. 

Author Response

REVIEWER #2:

COMMENT: The technical note entitled: Chemical analysis of plasma-activated culture media by ion 2 chromatography is well written, solid article which may be interest for Pharmaceuticals Journal readers.

RESPONSE: Thank you for your positive feedback and for recognizing the relevance of our work. We greatly appreciate your suggestions and will carefully address and implement all your recommendations to enhance the quality of the manuscript.

COMMENT: There are a few minor suggestions to be reconsidered, namely:

  1. Please add section described statistical tools and NCA software.

RESPONSE: Thank you for your suggestion. As requested, we have added a section describing the statistical tools and NCA software used in this study (page 8/12).

COMMENT: 2.  Please define validation procedures, including applied guidelines.

RESPONSE: Thank you for your comment. As requested, we have added a description of the validation procedures, including the applied guidelines, to the revised manuscript (page 4/12).

COMMENT: 3. Please add a conclusion section. A small discussion will be beneficial.

RESPONSE: Thank you for your valuable suggestion. We have added a conclusion section and included a brief discussion to enhance the clarity and comprehensiveness of the manuscript. We believe this addition strengthens the overall quality of the paper.

COMMENT: To sum up, I reccomend the publication after reconsideration of suggestions. 

RESPONSE: Thank you for your recommendation and constructive feedback. We have carefully addressed all the suggested revisions and believe the manuscript has been significantly improved as a result. We appreciate your support and look forward to the publication.

Reviewer 3 Report

Comments and Suggestions for Authors

In their Technical Note entitled Chemical analysis of plasma-activated culture media by ion chromatography Locatelli et al. discuss their notion, that colorimetric assays are inferior to ion chromatography with respect to the quantification of RONS produced in cell culture media by the exposure to cold atmospheric plasma (CAP).The line of argument is clear and convincing and it can be anticipated that many researchers interested in qunatifying ROS will find the suggested ion chromatography very attractive for this purpose.

I have only minor technical comments:

On page 2, line 58 in “ringer’s lactate solution”, an apostrophe should be used instead of single right quotation mark, I suppose.

On page 2, line 61-65 in the sentence “CAP demonstrated remarkable anti-tumoral potential across a wide range of cancer 61 types [5], although the underlying mechanisms are still not clear. Several types of short-62 lived (e.g. radical hydroxyl •OH, anion superoxide O2•−, and singlet oxygen 1O2) and 63 long-lived (hydrogen peroxide H2O2, nitrite NO2−, and nitrate NO3−) RONS have been 64 detected both in the gas and liquid phases [6].“ The typesetting oft he different chemical species must be corrected using e.g. subscript and superscript settings.

These corrections seem necessary throughut the text, e.g. for H2O2 (should read H2O2, of course).

1-(H)-naphthotriazole should read 1H-naphthotriazole (with italic H)

In Figure 2, sulphate should better be spelled sulfate.

Instead of “selective, accurate, rapid and sensible quantification“, I guess it should be selective, accurate, rapid and sensitive quantification“.

Comments on the Quality of English Language

The language of the manuscript should definitely be improved by increasing the frequency in the use of articles (a / the), e.g. in: „Griess test, used to detect stable metabolites of nitric oxide (NO), such as nitrites and nitrates, is a colorimetric assay with UV/Vis absorbance (maximum at 540 nm), based on the reaction with sulfanilic acid and N-(1-naphthyl)-ethylene diamine hydrochloride which generates a pink com-pound.” (The locant N in N-(1-naphthyl)-ethylene diamine should be italicized and in “the measurement of hydroxyl group (OH)“.

Names of chemical compounds are considered common nouns which are capitalized at the beginning of titles, subtitles, and sentences but not elsewhere. Thus, “reaction of 2,3-Diaminonaphtalene (DAN)” should be spelled “reaction of 2,3-diaminonaphtalene (DAN)”, “As for ROS quantification, 2′,7′-Dichlorodihydrofluorescein” should be spelled “As for ROS quantification, 2′,7′-dichlorodihydrofluorescein”, here, primes should be used instead of single right quotation marks, I suppose.

Please check the grammar of verbs like „show“, in cases like „(data not showed)“ and „Thus, with Excel we obtained the Pharmacokinetics parameters, previously showed in di Giacomo et al. [11, 12].“.

Author Response

REVIEWER #3:

COMMENT: In their Technical Note entitled „Chemical analysis of plasma-activated culture media by ion chromatography Locatelli et al. discuss their notion, that colorimetric assays are inferior to ion chromatography with respect to the quantification of RONS produced in cell culture media by the exposure to cold atmospheric plasma (CAP).The line of argument is clear and convincing and it can be anticipated that many researchers interested in qunatifying ROS will find the suggested ion chromatography very attractive for this purpose.

RESPONSE: Thank you for your thoughtful and positive feedback. We greatly appreciate your recognition of the clarity and relevance of our work. We will carefully follow your recommendations to further enhance the manuscript.

COMMENT: I have only minor technical comments: On page 2, line 58 in “ringer’s lactate solution”, an apostrophe should be used instead of single right quotation mark, I suppose.

RESPONSE: Thank you for pointing out this detail. We have corrected the issue by replacing the single right quotation mark with an apostrophe as suggested (page 2/12).

COMMENT: On page 2, line 61-65 in the sentence “CAP demonstrated remarkable anti-tumoral potential across a wide range of cancer 61 types [5], although the underlying mechanisms are still not clear. Several types of short-62 lived (e.g. radical hydroxyl •OH, anion superoxide O2•−, and singlet oxygen 1O2) and 63 long-lived (hydrogen peroxide H2O2, nitrite NO2−, and nitrate NO3−) RONS have been 64 detected both in the gas and liquid phases [6].“ The typesetting of the different chemical species must be corrected using e.g. subscript and superscript settings.

RESPONSE: Thank you for your observation. We have corrected the typesetting of the chemical species, using the appropriate subscript and superscript formatting, and have also unified the presentation of the chemical names throughout the manuscript.

COMMENT: These corrections seem necessary throughut the text, e.g. for H2O2 (should read H2O2, of course).

RESPONSE: Thank you for pointing this out. We have carefully reviewed the manuscript and corrected the formatting for chemical names throughout the text, ensuring consistency and proper representation, such as for H₂O₂.

COMMENT: 1-(H)-naphthotriazole should read 1H-naphthotriazole (with italic H)

RESPONSE: Thank you for your attention to detail. We have corrected the formatting of "1-(H)-naphthotriazole" to "1H-naphthotriazole" with the italicized "H" as suggested (page 2/12).

COMMENT: In Figure 2, sulphate should better be spelled sulfate.

RESPONSE: Thank you for your suggestion. We have corrected the spelling of "sulphate" to "sulfate" in Figure 2 as recommended.

COMMENT: Instead of “selective, accurate, rapid and sensible quantification“, I guess it should be “selective, accurate, rapid and sensitive quantification“.

RESPONSE: Thank you for your helpful suggestion. We have corrected the phrasing to "selective, accurate, rapid, and sensitive quantification" as recommended (page 4/12).

COMMENT: The language of the manuscript should definitely be improved by increasing the frequency in the use of articles (a / the), e.g. in: „Griess test, used to detect stable metabolites of nitric oxide (NO), such as nitrites and nitrates, is a colorimetric assay with UV/Vis absorbance (maximum at 540 nm), based on the reaction with sulfanilic acid and N-(1-naphthyl)-ethylene diamine hydrochloride which generates a pink com-pound.” (The locant N in N-(1-naphthyl)-ethylene diamine should be italicized and in “the measurement of hydroxyl group (OH)“.

Names of chemical compounds are considered common nouns which are capitalized at the beginning of titles, subtitles, and sentences but not elsewhere. Thus, “reaction of 2,3-Diaminonaphtalene (DAN)” should be spelled “reaction of 2,3-diaminonaphtalene (DAN)”, “As for ROS quantification, 2′,7′-Dichlorodihydrofluorescein” should be spelled “As for ROS quantification, 2′,7′-dichlorodihydrofluorescein”, here, primes should be used instead of single right quotation marks, I suppose.

RESPONSE: Thank you for your detailed and valuable suggestions. We have thoroughly reviewed the manuscript to improve the language, ensuring a more consistent and appropriate use of articles (a/the) throughout. Additionally, we have corrected the formatting of chemical compound names, italicized the locant "N" in N-(1-naphthyl)-ethylene diamine, and replaced single right quotation marks with primes where necessary. These adjustments align with your recommendations and enhance the clarity and accuracy of the text.

COMMENT: Please check the grammar of verbs like „show“, in cases like „(data not showed)“ and „Thus, with Excel we obtained the Pharmacokinetics parameters, previously showed in di Giacomo et al. [11, 12].“

RESPONSE: Thank you for bringing this to our attention. We have carefully reviewed and revised the manuscript to correct the grammatical usage of verbs such as "show," ensuring proper consistency and accuracy throughout the text.

Reviewer 4 Report

Comments and Suggestions for Authors

1.English needs to be revised and there are formatting isue and typos

2.some pictures regarding the intsrumentation, like schematics would be helpfull earlier in the text as well, to coampre the difference between the presented types in the introduction

3.a lot of editing of the text is required

The topis is in general interesting, but english is quite challenging and needs to be improved.

Comments on the Quality of English Language

Needs a lot of improvments

Author Response

REVIEWER #4:

COMMENT: 1. English needs to be revised and there are formatting isue and typos

RESPONSE: Thank you for your feedback. We have thoroughly revised the manuscript to improve the English language, address formatting issues, and correct any typos.

COMMENT: 2. some pictures regarding the intsrumentation, like schematics would be helpfull earlier in the text as well, to coampre the difference between the presented types in the introduction

RESPONSE: Thank you for your suggestion. We have added a schematic representation of the instrument used for this study earlier in the text to provide a clearer comparison of the different types mentioned in the introduction (page 6/12).

COMMENT: 3. a lot of editing of the text is required

RESPONSE: Thank you for your feedback. We have conducted a comprehensive revision of the manuscript, thoroughly editing the text to improve its clarity, coherence, and overall quality.

COMMENT The topis is in general interesting, but english is quite challenging and needs to be improved.

RESPONSE: Thank you for your feedback. We have thoroughly revised the manuscript to improve the English language, ensuring greater clarity and readability. We hope the updated version meets your expectations.
